# Insight into partial agonism by observing multiple equilibria for ligand-bound and $G_s$-mimetic nanobody-bound $\beta_1$-adrenergic receptor

Andras S. Solt[1], Mark J. Bostock [1], Binesh Shrestha[2], Prashant Kumar[1], Tony Warne[3], Christopher G. Tate [3] & Daniel Nietlispach [1]

A complex conformational energy landscape determines G-protein-coupled receptor (GPCR) signalling via intracellular binding partners (IBPs), e.g., $G_s$ and $\beta$-arrestin. Using $^{13}C$ methyl methionine NMR for the $\beta_1$-adrenergic receptor, we identify ligand efficacy-dependent equilibria between an inactive and pre-active state and, in complex with $G_s$-mimetic nanobody, between more and less active ternary complexes. Formation of a basal activity complex through ligand-free nanobody–receptor interaction reveals structural differences on the cytoplasmic receptor side compared to the full agonist-bound nanobody-coupled form, suggesting that ligand-induced variations in G-protein interaction underpin partial agonism. Significant differences in receptor dynamics are observed ranging from rigid nanobody-coupled states to extensive μs-to-ms timescale dynamics when bound to a full agonist. We suggest that the mobility of the full agonist-bound form primes the GPCR to couple to IBPs. On formation of the ternary complex, ligand efficacy determines the quality of the interaction between the rigidified receptor and an IBP and consequently the signalling level.

[1] Department of Biochemistry, University of Cambridge, 80 Tennis Court Road, Cambridge CB2 1GA, UK. [2] Protein Sciences, CBT, Novartis Institutes for BioMedical Research (NIBR), CH-4002 Basel Switzerland. [3] Medical Research Council Laboratory of Molecular Biology, Cambridge Biomedical Campus, Francis Crick Avenue, Cambridge CB2 0QH, UK. Correspondence and requests for materials should be addressed to D.N. (email: dn206@cam.ac.uk)

G-protein-coupled receptors (GPCRs) relay an extracellular stimulus across the plasma membrane, activating cellular signalling pathways via coupling to intracellular binding partners (IBP) such as heterotrimeric G-proteins and β-arrestins. GPCRs control a wide range of physiological processes and are implicated in many disease states[1]. Rhodopsin-like GPCRs are estimated to be targeted by ~30% of current drugs[2]. β-adrenergic receptors (βAR) are essential in the sympathetic nervous system; $\beta_1$AR is the predominant subtype in the heart, and is targeted by drugs such as β-blockers in the context of heart failure[3]. βARs show constitutive (basal) activity, with $\beta_1$ at a lower but nevertheless significant level, compared to $\beta_2$AR[4, 5].

Over 170 crystal structures have been solved of nearly 40 unique receptors[6, 7] in the presence of different ligands and IBPs[8–10], substantially developing our understanding of how these receptors function. NMR studies indicate a complex, dynamic conformational energy landscape consisting of several states in equilibrium[11–20], with substantial variations for different receptors. Allosteric coupling networks are observed[17] along with correlations of ligand structure and efficacy with spectral parameters[12, 17].

GPCRs are known to show varying degrees of constitutive activity[21], with sampling by the receptor in the absence of a ligand (apo form) of activated states observed in fluorescence[22] and NMR studies[18]. Addition of ligands shifts this equilibrium either towards the inactive form (inverse agonists) or towards activated conformations (agonists), which can couple to multiple intracellular signalling pathways[18, 23].

Crystallisation of a receptor in the fully active state requires the presence of a cytoplasmic signalling partner. A range of native binding partners and mimetics ($G_s$, nanobodies, β-arrestins, mini-$G_s$ and the transducin α-subunit C-terminal peptide)[10] have been used, and the resulting active structures for the μ-opioid receptor (μOR)[24], adenosine $A_{2A}$ receptor ($A_{2A}$R)[25], $\beta_2$AR[26, 27], muscarinic acetylcholine receptor M2[28] and rhodopsin[29] are all very similar pointing towards a unified mechanism for receptor activation. The hallmarks of an active state are a large lateral displacement of the cytoplasmic end of transmembrane helix 6 (TM6) from the helix bundle, accompanied by a characteristic rearrangement of conserved residues (Arg[3.50], Tyr[5.58] and Tyr[7.53] (superscripts refer to Ballesteros–Weinstein numbering[30])) that seem to stabilise the receptor in its active conformation[10]. For agonist-bound $A_{2A}$R, an intermediate-active state structure is found in the absence of IBP and there is also spectroscopic evidence for an active-like intermediate[19, 31]. No such equivalent has been observed for agonist-bound $\beta_1$AR, reflecting subtle variations in the conformational energy landscape. However, NMR studies of $\beta_1$- and $\beta_2$AR indicate a ligand efficacy-dependent equilibrium between inactive and active states[11, 12, 17]. In addition, accelerated molecular dynamics (MD) simulations may be used to assess receptor activation, e.g., following the pathway from the active (ternary) structure of $\beta_2$AR to the inactive receptor, potentially providing evidence for an intermediate conformation, as well as coupling between the ligand and G-protein-binding sites[32]. However, currently for $\beta_1$AR, there are no reported active state structures.

Evidence indicates that a range of inactive, intermediate, active-like and active states are involved in GPCR function, which likely exist in equilibrium. However, crystallographic studies are unable to address questions regarding the role of receptor dynamics accompanying the process of activation, and further questions remain related to agonist efficacy, partial agonism, constitutive activity and sampling of active states in the absence of G-protein.

Here, we use NMR to investigate the conformational diversity and dynamic nature of coupling between activated receptor states of turkey $\beta_1$AR and $G_s$-mimetic nanobodies. We show that $\beta_1$AR in an active ternary complex has differing spectral signatures when bound to diverse activating ligands that correlate ligand efficacy, providing structural insight into the mechanism of partial agonism. Structural variations on TM5 and TM6 point to differences in the receptor–IBP interaction, which we hypothesise regulates the receptor's guanine–nucleotide exchange factor (GEF) activity. Our study also indicates extensive conformational changes that affect large parts of the receptor during ligand-independent basal activation. We show that both the ternary complexes and the nanobody-only bound forms are less dynamic, while the full agonist-bound receptor in the absence of IBP is at its most dynamic form, indicating extensive sampling of different active-like conformational states that are primed to bind a range of intracellular signalling partners. This emphasises the vital role of receptor dynamics for coupling to a variety of different signalling pathways, for biased signalling and for partial agonism.

## Results

**$\beta_1$AR constructs and NMR signal assignment.** A thermo-stabilised turkey $\beta_1$AR[33] construct, modified from the previously published β44-m23 $\beta_1$AR[34] was functionally expressed in baculovirus-infected insect cells (Methods). Compared to the published construct, the level of thermostabilisation was substantially reduced through reverse mutagenesis of V90M[2.53], A227Y[5.58] and L282A (IL3). The final $\beta_1$AR construct (Supplementary Fig. 1) contained R68S[1.59], E130W[3.41] and F327A[7.37] as the only remaining thermostabilising mutations and C116L[3.27] and C358A (CT) for improved expression yields[17]. Selective [13]C methyl methionine labelling involved supplementing methionine-deficient SF4 growth media with labelled amino acid. To reduce overlap in the NMR spectra, the number of methionine residues was reduced from 12 (including the N-terminus) to seven through removal of M44L, M48L, M179L, M281A and M338A creating $\beta_1$AR-MetΔ5, with methionines M1, M90[2.53] (TM2), M153 (IL2), M178[4.62] (TM4/EL2), M223[5.54] (TM5), M283[6.28] (TM6) and M296[6.41] (TM6) remaining (Supplementary Fig. 1b). For some samples, L108M (EL1) and L190M (EL2) were introduced providing additional reporters in extracellular loops 1 and 2. None of these additional methionines altered the thermal stability of the receptor. A comparison with residues in other class A GPCRs is shown in Supplementary Table 1. The expressed receptor was solubilised in lauryl-maltose-neopentyl-glycol (LMNG) and purified to 95% homogeneity judged by SDS–PAGE, using one-step nickel affinity chromatography. The methyl methionine signals of [[13]C-methyl-Met] $\beta_1$AR-MetΔ5 were assigned in [1]H, [13]C SOFAST heteronuclear multiple quantum coherence (HMQC) spectra using a mutagenesis approach for a range of different ligand-bound states (Supplementary Fig. 2). In the case of agonistic ligands, this assignment approach was extended to the nanobody-bound ternary complexes and to the ligand-free nanobody-bound receptor form.

**Correlations between ligand efficacy and chemical shift.** To examine the effect of ligand binding, [1]H, [13]C HMQC spectra were recorded at 308 K for $\beta_1$AR-MetΔ5 in the apo form and in the presence of saturating amounts of 7-methylcyanopindolol (very weak partial agonist)[35], carvedilol (weak partial agonist), cyanopindolol (weak partial agonist), salbutamol (partial agonist), isoprenaline (full agonist) and adrenaline (full agonist), respectively (Supplementary Table 2)[36]. Apo- and ligand-bound spectra (Supplementary Fig. 3) showed substantial chemical shift changes for M223[5.54] and M296[6.41] (TM5, TM6), which correlated with the ligand efficacy (Fig. 1), suggesting an equilibrium involving two states exchanging fast relative to the chemical shift difference between the two signals, known as the NMR timescale

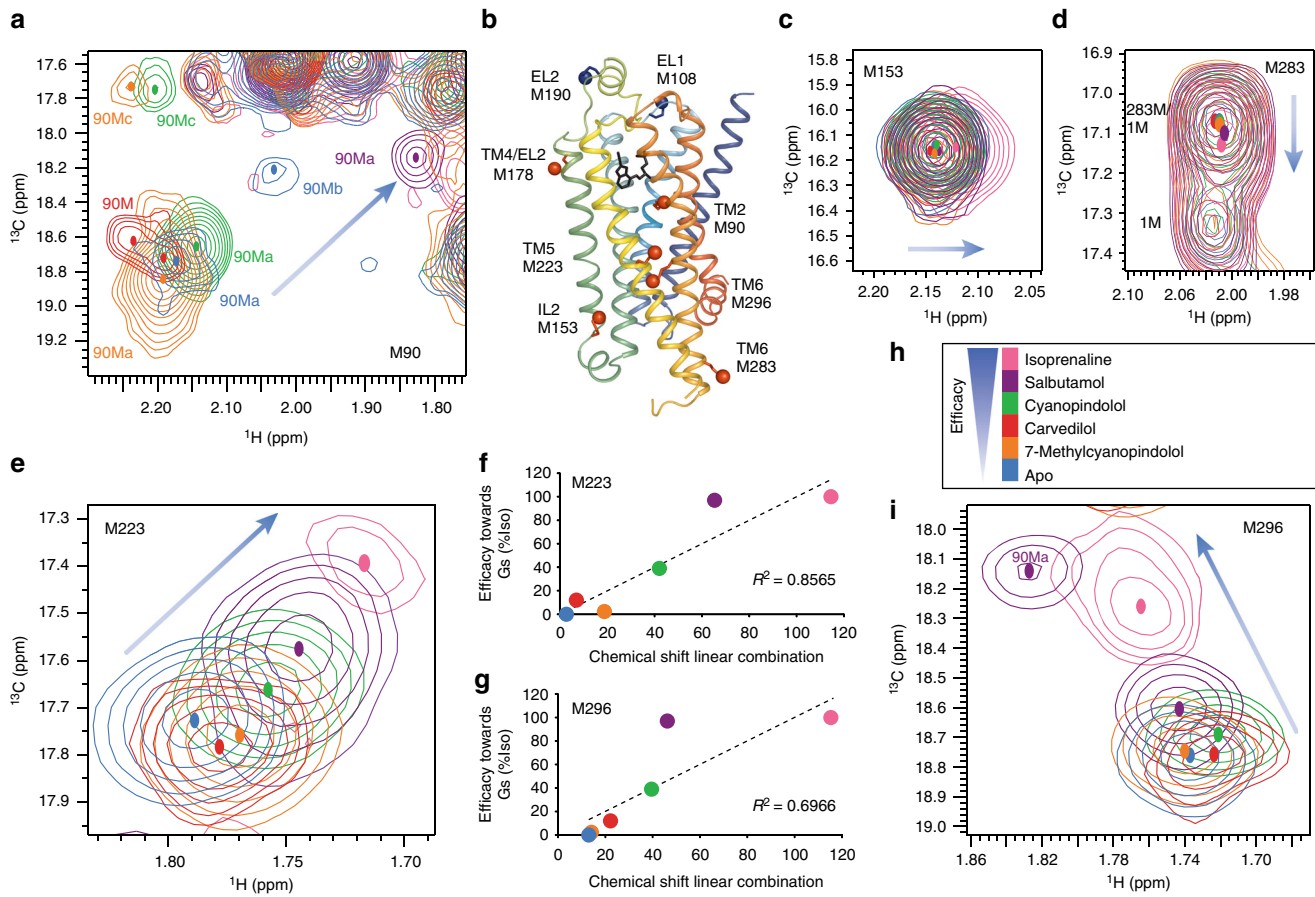

**Fig. 1** $^1$H, $^{13}$C HMQC spectra of ligand-bound methyl $^{13}$C-Met β$_1$AR-MetΔ5-L190M show resonance positions that correlate with ligand efficacies. Chemical shift changes for 2D $^1$H, $^{13}$C HMQC spectra for [$^{13}$C-methyl-Met] β$_1$AR shown for the apo form (blue) and orthosteric ligands in order of efficacy (**h**); 7-methylcyanopindolol (very weak partial agonist, orange), carvedilol (weak partial agonist, red), cyanopindolol (weak partial agonist, green), salbutamol (partial agonist, purple) and isoprenaline (full agonist, pink). The crystal structure of β$_1$AR (PDB code 4BVN) is shown in **b** with methionine residues used in this study shown as red spheres. The various methionines are shown separately; **a** M90, **c** M153, **e** M223, **d** M283, **i** M296. The centres of the resonances are indicated by coloured dots. Correlations between ligand efficacy and a linear combination of the chemical shifts (a$\delta_{1H}$ + b$\delta_{13C}$ + c) are shown in **f** and **g** with M223 fit to −1045.1$\delta_{1H}$ − 111.0$\delta_{13C}$ + 3839.8 and M296 to −634.1$\delta_{1H}$ − 239.9$\delta_{13C}$ + 5614.1. Full spectra are shown in Supplementary Fig. 3

(Supplementary Note 1). M223$^{5.54}$ and M296$^{6.41}$, located in the cytoplasmic half of the receptor, are sensitive reporters of the receptor's activation state. Smaller changes were observed for M283$^{6.28}$ and M153 (IL2) emphasising that structural changes related to ligand binding extend as far as the cytoplasmic side of the receptor and well beyond the ligand-binding pocket (Fig. 1).

In β$_1$AR, M90$^{2.53}$ is on the extracellular half of TM2 pointing towards the ligand-binding pocket. M90$^{2.53}$ samples a major and minor conformation in the unliganded state (M90a and M90b), indicative of at least two receptor conformations in the apo form (Fig. 1; Supplementary Fig. 3). Carvedilol-bound receptor showed two peaks for M90$^{2.53}$ at chemical shifts different from those in the apo form but close to the M90a state. For the partial agonists 7-methylcyanopindolol, cyanopindol and salbutamol, the major peak, which also corresponded to the major signal in the apo state, gradually shifted towards smaller ppm values. Notably, M90$^{2.53}$ was absent from the isoprenaline-bound spectrum, suggesting a more dynamic state in complex with a full agonist (Fig. 1; Supplementary Fig. 3). In addition, two shifts were observed for M178$^{4.62}$ (TM4) in all states, consistent with sampling of at least two conformations by the EL2 region (Supplementary Fig. 3).

**β$_1$AR shows increased dynamics in the agonist-bound state.** Loss of the M90$^{2.53}$ peak from the isoprenaline-bound β$_1$AR

spectra suggested increased signal broadening due to conformational exchange with rates similar to the NMR timescale (Supplementary Table 3). Short sample lifetimes and low experimental sensitivity limited the use of NMR spin relaxation tools, although these have been used for single-site measurements[37], while interpreting signal linewidths was also unreliable (Supplementary Note 2). Instead, we used relative and normalised peak intensities as a proxy for detecting varying protein dynamics between individual receptor forms. Intensity variations were particularly pronounced for M90$^{2.53}$, M223$^{5.54}$ and M296$^{6.41}$. M223$^{5.54}$ and M296$^{6.41}$ are sensitive reporters of the activation state on the cytoplasmic side of β$_1$AR and for each spectrum, their peak intensities were expressed as an intensity ratio relative to M153 (IL2) as a reference signal (Supplementary Table 4). Relative peak intensities for M223$^{5.54}$ and M296$^{6.41}$ were determined for two ligand series of β$_1$AR-MetΔ5 and β$_1$AR-MetΔ5-L190M, respectively (Supplementary Fig. 4). Smaller intensities indicated increasingly dynamic, exchange-broadened states. Normalised intensities were obtained by expressing the relative ratios as a proportion of the maximum peak intensity for a particular residue in a full series of spectra, where '1' represents the least dynamic state and smaller values indicate increased conformational fluctuations on the μs-to-ms timescale (Fig. 2). While relative intensities allowed a direct comparison of M223$^{5.54}$ and M296$^{6.41}$ conformational dynamics, normalised intensities

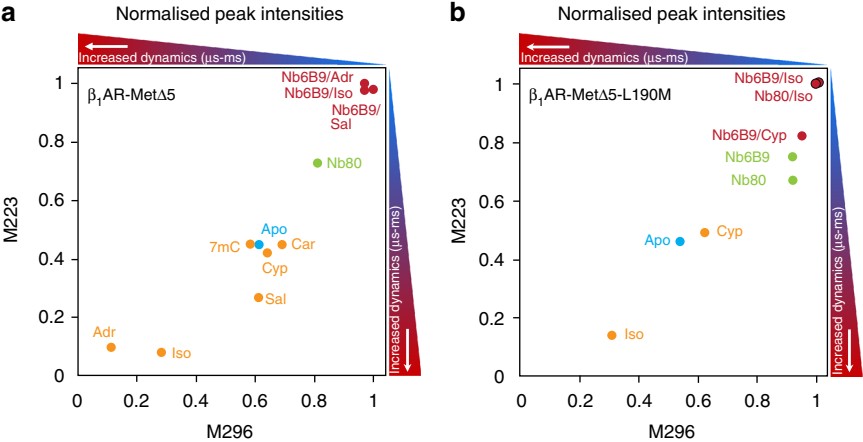

**Fig. 2** Normalised peak intensities of M223 and M296 reflect differences in dynamics that depend on the receptor state. Normalised peak intensities comparing residue M223 vs. M296 for the $\beta_1$AR-Met$\Delta$5 (**a**) and $\beta_1$AR-Met$\Delta$5-L190M (**b**) constructs. Relative intensities (Supplementary Fig. 4) were determined for each spectrum by calculating the peak intensity relative to the M153 signal. This was converted to a normalised intensity by setting the maximum intensity in each series to one, where one represents a lack of slow conformational fluctuations that lead to peak broadening and lower values represent increased µs-to-ms conformational dynamics. The different receptor states are shown as colour-coded circles: ligand bound (orange), apo (blue), nanobody bound (green) and ternary complex (red). The coloured bars along the top and the right-hand side of each graph indicate the direction of increasing µs–ms dynamics for residues M223 and M296, respectively. The red area indicates receptor states that are highly dynamic (µs–ms timescale) and blue indicates less dynamic, more rigid receptor states. Both constructs show a very similar pattern of intensities (dynamics) with full agonist-bound forms showing the greatest mobility for M223 and M296 and ternary complexes showing the greatest restriction in motion for these residues. Ligand-bound receptor and the apo form cluster around 0.5–0.6 normalised intensity units indicating some amount of motion (for a discussion of timescales see Supplementary Notes 1 and 2)

allowed comparison of the overall dynamics between the different receptor states and constructs. To eliminate bias from the reference signal, intensity ratios were also determined using the M190 (EL2) peak and the methyl signal of receptor-bound LMNG as the reference signal. Similar trends were obtained for all three cases (Supplementary Fig. 5a; Supplementary Table 5).

The full agonist-bound forms with isoprenaline or adrenaline showed consistently lower relative intensities for residues M223[5.54] and M296[6.41] than the partial agonist-bound forms (salbutamol, cyanopindolol and 7-methylcyanopindolol) or the apo state. Salbutamol-bound $\beta_1$AR is more mobile than the other partial agonist-bound receptors but is still considerably more rigid than full agonist-bound receptor (Fig. 2). Very similar trends for the dynamics of the cytoplasmic side of TM5 and TM6 were also found for other mutants of $\beta_1$AR-Met$\Delta$5 (M178A, M223L, M283A and M296A) used for the residue-specific assignment of the methionine signals (Supplementary Fig. 6; Supplementary Table 4). The data consistency across a wide range of mutants and ligands supports the interpretation of signal intensities in the context of changes in receptor dynamics. However, other sources that influence $R_2$ relaxation, e.g., variations in dipolar interactions following side chain repacking could also affect peak heights.

At 308 K, M223[5.54] and M296[6.41] signals for isoprenaline-bound $\beta_1$AR-Met$\Delta$5-L190M were very weak. On lowering the temperature to 298 K, both signals increased in intensity, and a second signal for M223[5.54] was resolved (Fig. 3), characteristic of intermediate exchange on the NMR timescale at 308 K, leading to extensive peak broadening; upon temperature reduction, the exchange became slow enough to resolve the exchanging states as individual signals (Supplementary Table 3). Further temperature reduction to 288 K preserved the more intense main signal compared to 308 K; however, it became difficult to determine the total number of signals as some apparent peaks were very close to the noise level. Both M223[5.54] and M296[6.41] [13]C chemical shift positions shifted towards the apo-receptor peak as the temperature was decreased (Supplementary Fig. 7; Supplementary Note 3).

**Structural and dynamic changes in the ternary complex**. To investigate conformational changes on forming the $\beta_1$AR ternary complex, we recorded [1]H, [13]C HMQC spectra of isoprenaline-bound $\beta_1$AR-Met$\Delta$5-L190M after the addition of a stoichiometric amount of nanobody, using two nanobodies, Nb80[26] and Nb6B9[38]. In the ternary full agonist/receptor/nanobody complexes, both nanobodies are known to have nM-binding affinities to the receptor[38]. Substantial changes were observed upon formation of the fully active ternary complex through addition of Nb6B9 (Fig. 4). Peak positions for residues in the distant extracellular region were affected as well as in the cytoplasmic region proximal to the nanobody-binding site. The largest chemical shift changes were observed for M223[5.54] and M296[6.41] indicating substantial changes in TM5 and TM6 upon nanobody docking. Large displacements on the cytoplasmic side were also seen for M283[6.28] (TM6) and M153 (IL2). The changes were much more substantial than for full agonist binding to the apo form (without nanobody) (Supplementary Fig. 8). Differences were also detected on the extracellular side for M178[4.62] (TM4/EL2), M190[5.21] (EL2) and M108 (EL1) (Fig. 4). Identical results were obtained for binding to Nb80 (Supplementary Fig. 2p).

Compared to ligand-only-bound receptor, normalised peak intensities for M223[5.54] and M296[6.41] were substantially increased in the ternary complex, indicating significantly more rigid receptor on the µs-to-ms timescale (Fig. 2). This was consistently observed for both nanobodies with $\beta_1$AR-Met$\Delta$5-L190M as well as for other $\beta_1$AR-Met$\Delta$5 mutant receptor constructs (M153A, M178A, M223L, M283A and M296A) (Supplementary Fig. 6).

**Ligand efficacy affects ternary complex chemical shifts**. The structural fingerprint of $\beta_1$AR ternary complexes was investigated in the presence of the partial agonist cyanopindolol, agonist salbutamol and full agonist adrenaline and the spectral signatures compared with the isoprenaline-bound ternary complex, using the higher-affinity nanobody, Nb6B9. Changes were observed in both the intracellular and extracellular halves of the GPCR, with

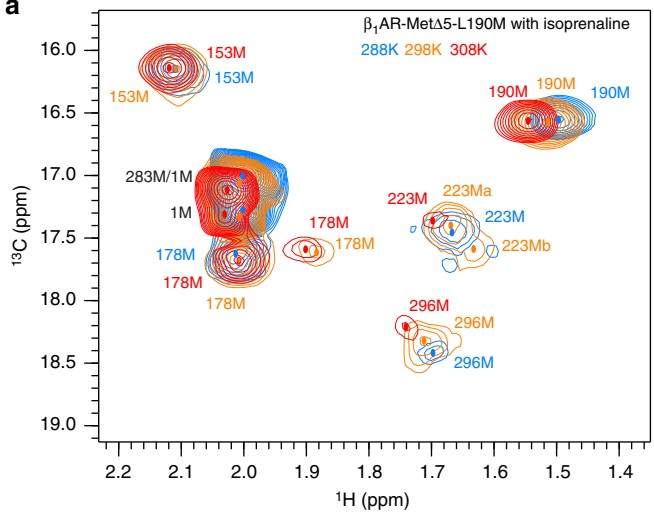

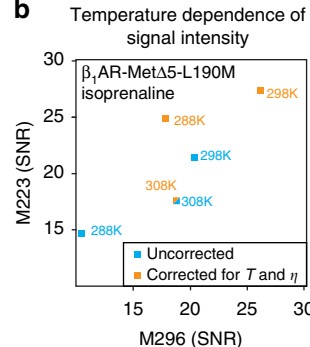

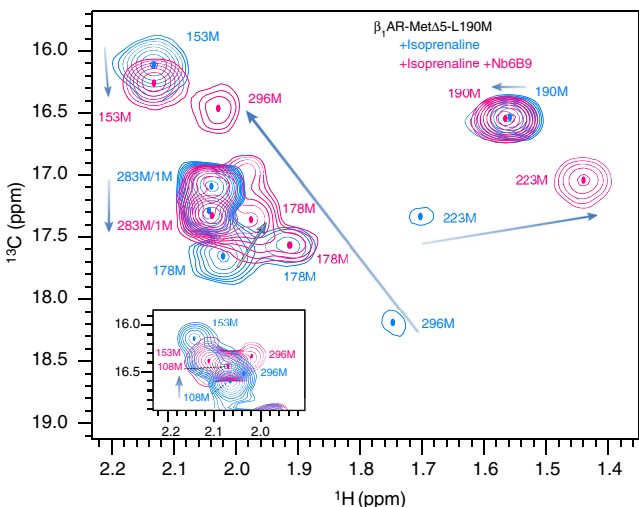

**Fig. 4** Changes in resonance positions upon formation of the isoprenaline-bound $\beta_1$AR-Met$\Delta$5-L190M ternary complex with a nanobody. 2D $^1$H, $^{13}$C HMQC spectra for [$^{13}$C-methyl-Met] $\beta_1$AR-Met$\Delta$5-L190M showing substantial chemical shift changes to isoprenaline-bound $\beta_1$AR (blue) on addition of Nb6B9 nanobody (pink). Major changes from the isoprenaline-bound receptor to the ternary complex are indicated by arrows. The inset shows changes for M108 as observed in EL1 of $\beta_1$AR-Met$\Delta$5-L108M

**Fig. 3** Temperature dependence of chemical shifts and signal intensities in isoprenaline-bound $\beta_1$AR-Met$\Delta$5-L190M reveal a highly dynamic receptor. 2D $^1$H, $^{13}$C HMQC spectra (**a**) for the $\beta_1$AR-Met$\Delta$5-L190M construct recorded at 288 (blue), 298 (orange) and 308 K (red) showing the temperature dependence of the spectra. Resonance centres are indicated by coloured dots. All residues are seen to move with changing temperature, with substantial shifts for L190M and M1/M283, M223 and M296. Intensity changes are observed for M223 and M296 and at 298 K, M223 appears in two conformations (M223a and M223b). (Spectra were aligned using the isopropyl methyl group of unbound isoprenaline as a reference.) In **b** intensities, shown as signal-to-noise ratios (SNR), are plotted for M223 vs. M296 with uncorrected values in blue and corrected values accounting for the temperature and viscosity ($\eta$) changes (Supplementary Table 6) shown in orange. SNRs increase from 308 to 298 K indicating that at 308 K, M223 and M296 are in intermediate exchange leading to extensive peak broadening with a shift to slower exchange at lower temperatures, allowing separate states to be resolved as seen for M223 in **a**

dependence suggested that M223$^{5.54}$ and M296$^{6.41}$ existed in equilibrium between states exchanging faster than the NMR timescale (Supplementary Table 3). For the salbutamol-bound ternary complex, the decrease in temperature caused a reduction in relative peak intensities for M223$^{5.54}$ and M296$^{6.41}$ consistent with fast exchange becoming comparable to the NMR timescale (Supplementary Table 6).

**Ligand-free basal activity complex with a nanobody.** In view of the reported basal activity for $\beta$AR[5], structural evidence for an interaction between $\beta_1$AR and G-protein mimetic nanobody in the absence of an agonist was sought. The nanobody–$\beta_1$AR affinity was expected to be lower in the absence of agonists. Using a 15-fold nanobody excess, ligand-free receptor was observed predominantly as nanobody-bound (based on NMR signal integration: Nb6B9 99.5% bound; Nb80 96% bound). Substantial chemical shift changes were observed in the basal complex for both nanobodies relative to the apo form. Affected positions included residues at the extracellular side (M190 (EL2), M178$^{4.62}$) and at the cytoplasm (M223$^{5.54}$, M296$^{6.41}$, M283$^{6.28}$ and M153 (IL2)) (Fig. 6). Several residues showed slow exchange between the apo- and nanobody-bound receptor form (Supplementary Table 3), e.g., M223$^{5.54}$ and M296$^{6.41}$, allowing $K_d$ estimation using peak volumes. NMR titrations for $\beta_1$AR-Met$\Delta$5 and $\beta_1$AR-Met$\Delta$5-L190 in the apo form with Nb80 and Nb6B9 provided $K_d$ estimates of ca. 56 and 8 $\mu$M, respectively, which were several orders of magnitude weaker than in the presence of isoprenaline[38].

For the receptor bound to Nb80 or Nb6B9, relative signal intensities for M223$^{5.54}$ and M296$^{6.41}$ were smaller than in the ternary agonist complexes but noticeably larger than in the apo- or ligand-bound receptor forms (Fig. 2; Supplementary Fig. 4).

Overall peak positions for the basal activity complex appeared more similar to the ternary complexes than the apo- or isoprenaline-bound receptor (Supplementary Fig. 11). Further small shifts on addition of isoprenaline resulted in a spectrum that was superimposable with the ternary complex formed by addition of a nanobody to isoprenaline-bound receptor (Supplementary Fig. 12). Isoprenaline addition affected residues

substantial shifts detected for M178$^{4.62}$, and M296$^{6.41}$ and smaller shifts for M153 (IL2), M223$^{5.54}$ and M283$^{6.28}$ (Fig. 5; Supplementary Fig. 9).

Notably, M296$^{6.41}$ and to a lesser extent M153 (IL2), M178$^{4.62}$ and M223$^{5.54}$ showed a correlation between ligand efficacy and chemical shift positions (Fig. 5). As for nanobody-free receptor, M178$^{4.62}$ shows two conformations, with only one showing an efficacy-dependent shift. While these measurements were done at 308 K, a repeat of the measurement for the salbutamol-bound $\beta_1$AR-Met$\Delta$5-L190M ternary complex at 293 K showed a small shift change for M223$^{5.54}$ and M296$^{6.41}$ away from the signal of the ternary isoprenaline towards smaller ppm (Supplementary Figs. 7, 10). Comparable observations are also made for the cyanopindolol- and isoprenaline-bound complexes. The shift changes correlating with ligand efficacy together with the observed temperature

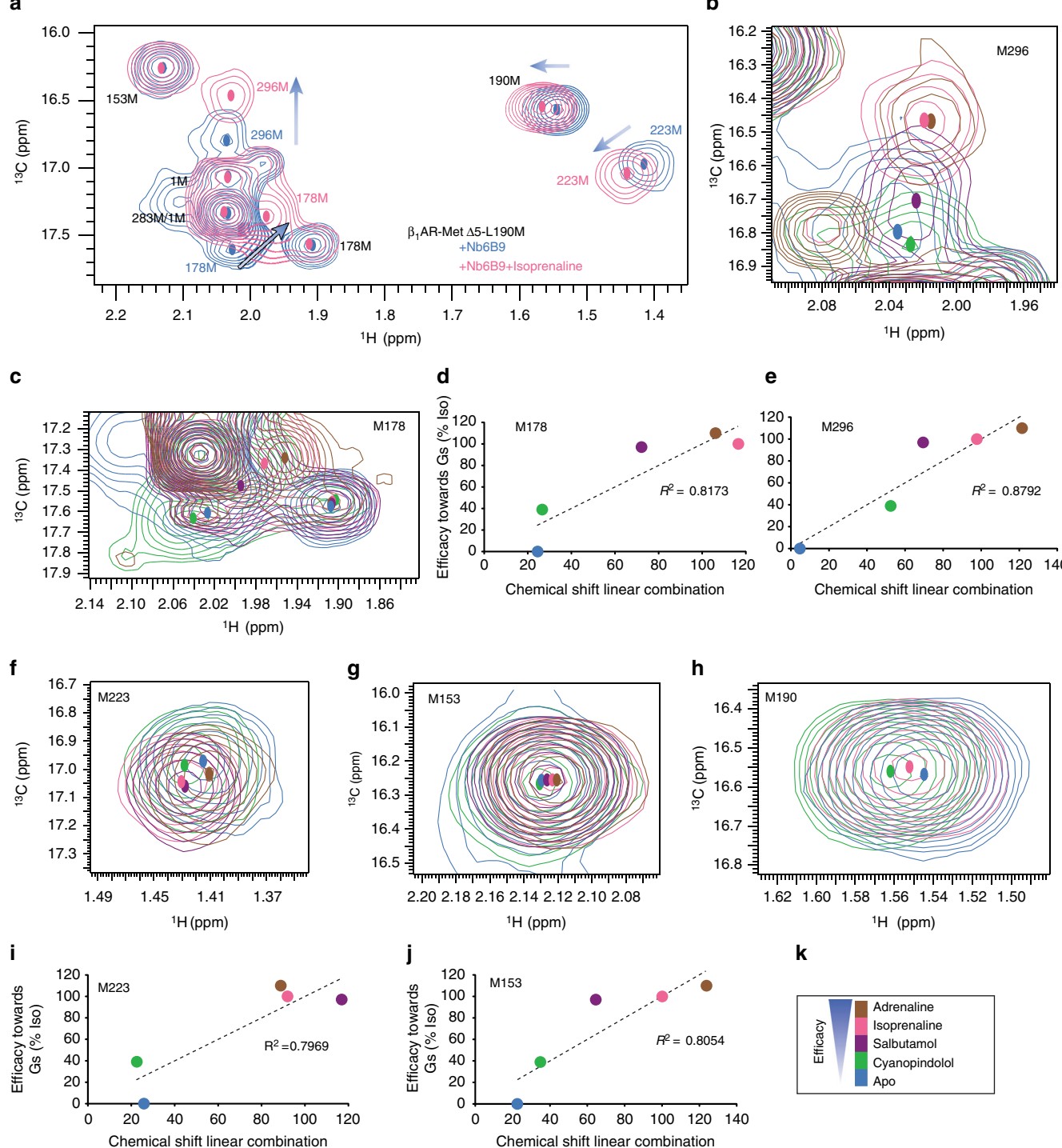

**Fig. 5** Resonance positions of ternary agonist complexes of β₁AR-MetΔ5-L190M with a nanobody correlate with ligand efficacies. 2D ¹H, ¹³C HMQC spectra for [¹³C-methyl-Met] β₁AR-MetΔ5-L190M or [¹³C-methyl-Met] β₁AR-MetΔ5 bound to Nb6B9 in the apo form (blue) or in a ternary complex with different orthosteric ligands; cyanopindolol (weak partial agonist, green), salbutamol (partial agonist, purple), isoprenaline (full agonist, pink) and adrenaline (full agonist, brown), in order of increasing ligand efficacy (**k**). The full spectrum for the apo form bound to Nb6B9 and full agonist-(isoprenaline) bound ternary complex is shown in **a**. Selected regions are also shown to illustrate the changes for residues M296 (**b**), M178 (**c**), M223 (**f**), M153 (**g**), M190 (**h**). Under each spectrum, a linear fit of ligand efficacy to a linear combination of the chemical shifts, $a\delta_{1_H} + b\delta_{13_C} + c$, is shown. The equations determined were as follows: M178 (**d**), $1510.6\delta_{1_H} - 732.7\delta_{13_C} + 9863.4$; M296 (**e**), $-5936.4\delta_{1_H} + 5.93\delta_{13_C} + 11985.7$; M223 (**i**), $-1607.4\delta_{1_H} + 1258.4\delta_{13_C} - 19056.1$ and M153 (**j**), $-11854.9\delta_{1_H} + 2691.2\delta_{13_C} - 18476.6$. Due to limited sample availability, two β₁AR constructs were employed: β₁AR-MetΔ5-L190M was used here with isoprenaline, cyanopindolol and in the apo form, while β₁AR-MetΔ5 was used with salbutamol and adrenaline. Except for the absence of the M190 signal in the spectra of the latter, the two β₁AR constructs were identical in terms of stability and binding properties and hence were used interchangeably

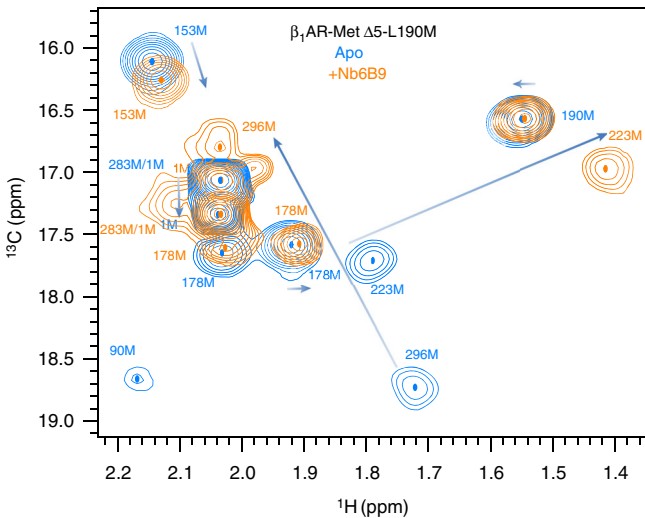

**Fig. 6** Formation of a ligand-free basal activity complex results in large changes in resonance positions. 2D $^1$H, $^{13}$C HMQC spectra for [$^{13}$C-methyl-Met] β$_1$AR-MetΔ5-L190M showing substantial chemical shift changes to the apo form of β$_1$AR (blue) on addition of Nb6B9 nanobody (orange). Major changes are indicated by arrows

including M108 (EL1), M178 (EL2) and M190 (EL2) on the extracellular side of the receptor and M223$^{5.54}$ and M296$^{6.41}$ on the cytoplasmic side. No changes were observed for M283$^{6.28}$. Similar observations were made when adding cyanopindolol, salbutamol or adrenaline to Nb6B9-bound β$_1$AR-MetΔ5.

**Increased activation barrier for thermostabilised M90A.** To further assess the influence of conformational dynamics on receptor activation, we introduced an additional thermostabilising mutation, M90A (β$_1$AR-MetΔ5-M90A), known to increase $T_m$ for the alprenolol-bound (antagonist) β$_1$AR$_{34-324}$ by +8 °C[39]. M296$^{6.41}$ signals for the isoprenaline- and salbutamol-bound states of the M90A mutant were slightly shifted towards the β$_1$AR-MetΔ5 signals when bound to lower-efficacy ligands (Fig. 7a). Similar shift changes were also observed for M223$^{5.54}$ with β$_1$AR-MetΔ5-M90A bound to salbutamol. Conversely, signal positions in the corresponding apo form were almost unchanged (Supplementary Fig. 13). The ligand-only-bound form (salbutamol and isoprenaline) of M90A gave substantially higher intensity values for M223$^{5.54}$ and M296$^{6.41}$, indicating a more rigid and less exchange-broadened receptor following the additional thermostabilisation. The differential increase in relative intensities followed the order apo < salbutamol < isoprenaline, with little change for the apo form compared to β$_1$AR-MetΔ5 (Fig. 7b). The relative signal intensities at 293 and 308 K showed no temperature dependence for M296$^{6.41}$ and M223$^{5.54}$ in the isoprenaline-bound state nor any field dependence (800 vs. 600 MHz), supporting a conformationally less dynamic, more homogeneous thermostabilised receptor (Supplementary Table 7a).

Isoprenaline-bound M90A ternary complexes show a similar overall conformational signature to β$_1$AR-MetΔ5 ternary complexes, with similar relative signal intensities (Fig. 7b; Supplementary Table 7b). However, the M223$^{5.54}$ and M296$^{6.41}$ signals of M90A were slightly shifted towards the partial agonist ternary complexes of β$_1$AR-MetΔ5 (Fig. 7c).

## Discussion

GPCRs require an IBP to populate their fully active state[24–29], characterised by a large translation of the cytoplasmic end of

TM6 compared to the agonist-bound state and the rearrangement of several key residues believed to stabilise the active state[10, 40]. GPCR activation is now understood within a complex conformational energy landscape, supported by recent NMR studies underpinning their highly dynamic nature[11–14, 18–20]. However, no active state structure of β$_1$AR currently exists, and many questions remain surrounding the detailed mechanism of activation, e.g., how partial agonism manifests through G-protein interaction.

Using $^{13}$C methyl methionine NMR, we investigated a minimally thermostabilised turkey β$_1$AR receptor. We studied apo- and ligand-bound states and their interaction with G-protein mimetic nanobodies (Nb80 and Nb6B9), assessing the dynamic nature of the receptor using a normalised intensity ratio for residues M223$^{5.54}$ and M296$^{6.41}$, which are located on the cytoplasmic side of TM5 and TM6 (Supplementary Note 2).

Ligand-bound β$_1$AR spectra show considerable shift changes for residues near the ligand-binding pocket, e.g., M90$^{2.53}$ and on the cytoplasmic side of the receptor, e.g., M153 (IL2), M223$^{5.54}$, M283$^{6.28}$ and M296$^{6.41}$ (Fig. 1; Supplementary Fig. 3). M90$^{2.53}$ shows multiple signals, with the major peak progressing to lower ppm values as ligand efficacy increases, comparable to M82$^{2.53}$ in β$_2$AR (M90a, Fig. 1)[12]. However, M82$^{2.53}$ in β$_2$AR[12] shows a more correlated response to ligands that alter the population of the two states seen in the apo form and a third conformation observed in the presence of a full agonist[12, 13]. In contrast, M90$^{2.53}$ shows two conformations (M90a and M90c) for 7-methylcyanopindolol- and cyanopindolol-bound receptor, while apo receptor shows a different minor conformation, M90b, and carvedilol-bound receptor a second peak close to the M90a position. The salbutamol-bound form likely represents a state sampling more of the active conformation. No clear change in populations is seen with ligand efficacy. Although M82$^{2.53}$ is observed in formoterol-bound (full agonist) β$_2$AR, other signals, e.g., M215$^{5.54}$ (M223$^{5.54}$ in β$_1$AR) are missing, indicating increased motion in the full agonist-bound state consistent with signal loss in the isoprenaline-bound form of β$_1$AR[12]. MD simulations for β$_2$AR suggest that sterically unhindered exchange between different methyl side chain rotamer environments takes place on the ns timescale[13], too fast to cause the observed multiple peaks and broadening, suggesting that the vicinity of the ligand-binding pocket in β$_1$AR is in equilibrium between multiple slowly exchanging states with a substantial increase in intermediate exchange for isoprenaline-bound receptor.

M223$^{5.54}$ and M296$^{6.41}$ undergo large chemical shift changes correlating with ligand efficacy, indicating a conformational equilibrium in fast exchange (Supplementary Table 3) between the inactive state (I), unable to bind a G-protein, and the pre-active state (A), a more dynamic conformation with increased accessibility on the cytoplasmic side, populated in the full agonist-bound receptor (Fig. 1; Supplementary Note 3). The low basal activity of β$_1$AR indicates that the apo form is almost entirely in the inactive state (I) with the cytoplasmic half of the receptor in the closed conformation. Spectral changes for M153 (IL2) and M283$^{6.28}$, although smaller, are further evidence of orthosteric long-range effects that extend as far as the cytoplasmic face of the receptor.

Correlation with ligand efficacy was observed in a backbone valine $^{15}$N NMR study of tsβ$_1$AR for Val 226$^{5.57}$, although no strong correlation was observed for residues in TM6 (Val 280$^{6.25}$ and Val 298$^{6.43}$)[17], possibly reflecting the increased sensitivity of side chain chemical shifts towards changes in conformational packing or the higher thermostabilisation of tsβ$_1$AR which increases conformational rigidity. Notably, ligand-dependent changes in TM6 were observed in β$_2$AR via the equivalent Met279$^{6.41}$ reporter[12].

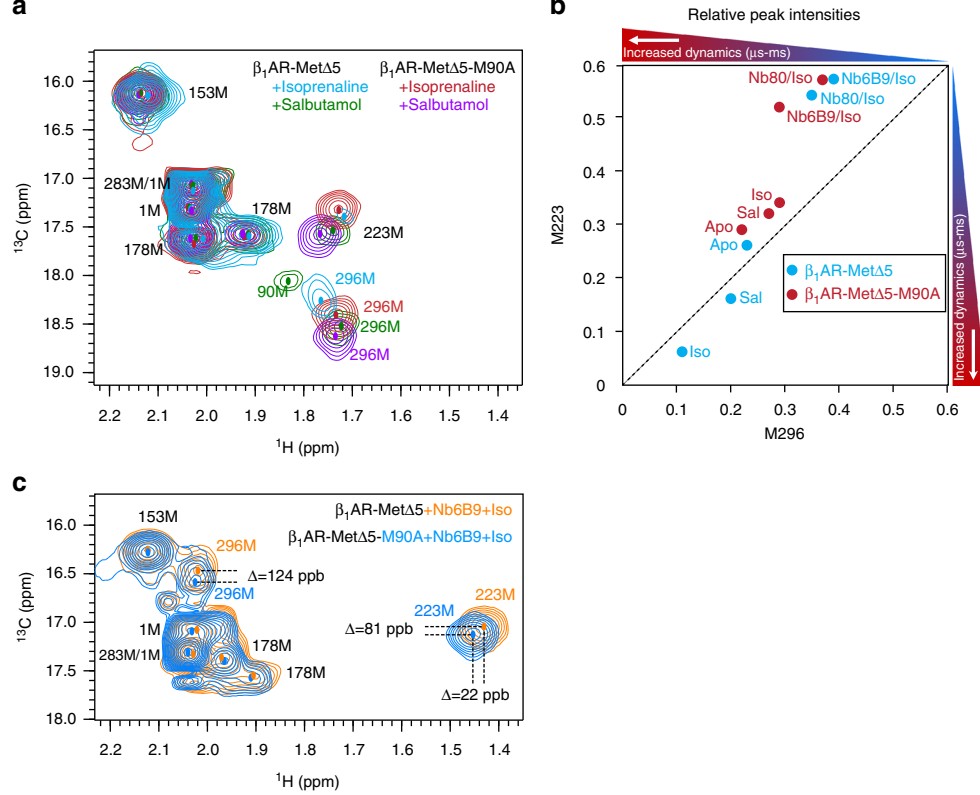

**Fig. 7** Additional thermostabilisation by M90A mutation shows β1AR-MetΔ5-M90A resonance positions and receptor dynamics characteristic of a less active receptor. 2D $^1H$, $^{13}C$ HMQC spectra (**a**) for [$^{13}C$-methyl-Met] β1AR-MetΔ5 (blue, green) and β1AR-MetΔ5-M90A (red, purple) in the ligand-bound form showing chemical shift changes with the orthosteric ligands isoprenaline (blue, red) and salbutamol (green, purple). Relative peak intensities (**b**) for ligand-bound and ternary complexes are shown for β1AR-MetΔ5 (blue) and β1AR-MetΔ5-M90A (red). The colour-coded bars at the top and the right-hand side of the peak intensity chart indicate the direction of increasing dynamics with generally less dynamic, rigid receptor states in blue, while red indicates receptor states which are increasingly dynamic on the μs-to-ms timescale. Compared to β1AR-MetΔ5, the M90A mutant shows more restricted dynamics in the ligand-only-bound forms, most notably for isoprenaline, indicating reduced conformational flexibility in this state. 2D $^1H$, $^{13}C$ HMQC spectra (**c**) of the ternary isoprenaline/Nb6B9-bound β1AR-MetΔ5 (orange) and β1AR-MetΔ5-M90A (blue) show overall a similar spectral fingerprint but with distinct differences in the positions of M223 and M296 that are shifted towards the positions found in partial agonist-bound ternary complexes with the less stabilised β1AR-MetΔ5, i.e., towards (A$^{G-}$)

Comparing normalised signal intensities for M223$^{5.54}$ and M296$^{6.41}$ shows that the full agonist-bound receptor signals are considerably weaker (~20% maximum intensity) than the other ligand-bound states and the apo form (~50% maximum intensity), indicating substantial broadening of the (A) state by μs-to-ms timescale exchange, with adrenaline-bound β1AR appearing even more dynamic than isoprenaline (Fig. 2; Supplementary Fig. 4). Lowering the temperature increases the intensity of these broad isoprenaline-bound receptor signals dramatically (Fig. 3a, b; Supplementary Fig. 7), suggesting a second, slower exchange process in the pre-active (A) state which is comparable to the NMR timescale, indicating sampling of multiple conformations by the (A) state. At 298 K, M223$^{5.54}$ appears as two resolved signals, indicating several conformations in slow exchange (Supplementary Table 3).

Overall, our data suggest that the pre-active state (A) is highly mobile on a μs-to-ms timescale, both near the ligand-binding pocket and on the cytoplasmic side of the receptor. This indicates that the cytoplasmic side samples at least two, and possibly more active-like states (A = A′, A″, A‴…) in intermediate exchange on the NMR timescale at 308 K, leading to strong signal broadening. These states may couple to different IBPs, in which case the ability of the receptor to access different conformations in the pre-active state plays an important role in engaging different downstream-signalling pathways. When bound to lower-efficacy

ligands or in the apo form, β1AR remains conformationally dynamic, however, less than in the full agonist-bound state. As the I ⇌ A equilibrium is shifted towards the more rigid (I) state, the receptor is conformationally less mobile as revealed by our intensity data (Fig. 2). A change in the lifetime of the (A = A′, A″, A‴…) states in the presence of different ligands is of course also possible.

Nanobodies, developed as a Gα_s protein mimic[26], were shown to interact with β1AR[41], emulating the effect of Gα_s-protein on the receptor and stabilising its active state. Isoprenaline-bound Nb80- or Nb6B9-β1AR-MetΔ5-L190M ternary complexes show dramatic spectral changes in the methionine region relative to isoprenaline-bound receptor (Fig. 4). The largest chemical shift changes are observed for M223$^{5.54}$ and M296$^{6.41}$, consistent with substantial structural rearrangements in TM5 and TM6 upon opening of the cytoplasmic half, as shown in β2AR crystal structures in complex with Nb6B9 or full G-protein[26, 27]. Methionine $^{13}C_ε$ chemical shifts report on $χ^3$ dihedral angles, indicating that for ligand-bound and apo states of β1AR, M296$^{6.41}$ is in the *trans* conformation and M223$^{5.54}$ shows *trans/gauche* exchange, shifting towards *gauche* on increased ligand efficacy, consistent with increased dynamics in the full agonist-bound form and the rotamer orientations in β2AR crystal structures (Supplementary Table 8). β2AR crystal structures were used due to bias in the existing β1AR structures towards antagonist-bound

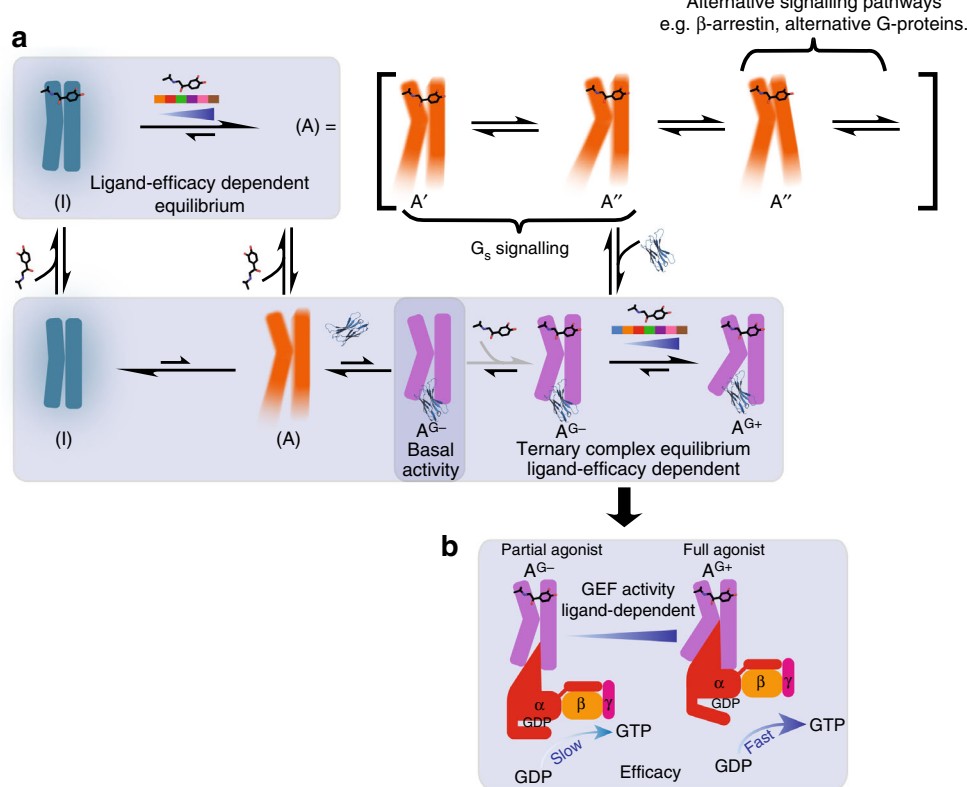

**Fig. 8** Model of $\beta_1$AR receptor activation showing partial agonism and basal activity. **a** $\beta_1$AR fluctuates between conformations that vary in the accessibility of their cytoplasmic side. Ligand binding determines the equilibrium position between a closed, inactive state (I) and a less occluded, activated form (A) of the receptor and also regulates the position of a second equilibrium with the active receptor bound to $G_s$. Agonist binding leads to a highly dynamic (A) state of the receptor that samples multiple active-like conformations (A′, A′′, A′′′...), which vary in their ability to interact with IBPs. Coupling to $G_s$ via these states leads to a more, $A^{G+}$, and a less active form, $A^{G-}$, which are in equilibrium with each other, with their relative populations influenced by the properties of the ligand bound. In the presence of a full agonist, both of these equilibria are shifted predominantly towards the (A) and in the presence of $G_s$, $A^{G+}$ states. In contrast, when bound to inactivating ligands, or in the apo form, the receptor predominantly populates the (I) state. In the apo form, small populations of the (A) state in the presence of $G_s$ can lead to the formation of the less active $A^{G-}$ state, consistent with basal activity at a low level. The basal activity form is subsequently able to bind a ligand (grey arrow), which can lead to a further shift of the active equilibrium towards the $A^{G+}$ state. While our in vitro data show the formation of the ternary complexes via this route (grey arrow) (Supplementary Fig. 12), the physiological significance of this pathway is unclear and the main contribution towards ternary complex formation is likely to occur via initial binding of a ligand, followed by IBP docking (arrows in black). **b** The more and less active forms of the coupled receptor show conformational differences on TM5 and TM6 that modulate their ability to interact with $G_s$. This could provide a mechanism to regulate signalling via the receptor's GEF function affecting the rate of GDP release. In the presence of partial agonists, a shift in equilibrium in the direction of $A^{G-}$ results in reduced levels of receptor signalling and lower efficacy

conformations, with limited changes in the cytoplasmic region following thermostabilisation (Supplementary Note 3; Supplementary Table 9)[34, 42]. On nanobody binding, M296[6.41] shifts substantially towards the *gauche* conformation correlating with increased ligand efficacy. The *gauche* rotamer conformation also increases slightly for M223[5.54] (Supplementary Note 3; Supplementary Fig. 14). As expected by their immediate proximity to the bound nanobody, residues M153 (IL2) and M283[6.28] also show large shift changes.

Spectral changes between isoprenaline-bound receptor and the corresponding ternary complex are much more dramatic than those between apo- and isoprenaline-bound $\beta_1$AR, suggesting that the (A) state corresponds to a pre-active rather than a fully active conformation (Supplementary Fig. 8). Notably, binding of a nanobody to the cytoplasmic side of the receptor transmits changes to the extracellular side of the receptor ('outward' effect), while binding of a full agonist in the orthosteric ligand-binding pocket results in changes as far as the cytoplasmic side of the receptor ('inward' effect) (Figs. 1, 4; Supplementary Fig. 8). Nanobody-induced changes at the extracellular side of the ternary complex include M108 (EL1), M178[4.62] (TM4/EL2) and M190

(EL2), suggesting an allosteric coupling network linking opposite faces of the receptor, where both the inward and outward effects reinforce ternary complex formation. An increase in agonist affinity frequently accompanies IBP binding and the observed conformational changes in EL1 and EL2 may be spectroscopic evidence for further contraction of the orthosteric ligand-binding environment[26, 27, 41].

The structural fingerprint of Nb6B9 ternary complex formation was further investigated for other ligands (Supplementary Fig. 9). Spectra show large chemical shift changes for M223[5.54] and M296[6.41], characteristic of the cytoplasmic opening of the receptor and nanobody docking, as previously observed with isoprenaline. However, substantial shift differences in the ternary complexes compared to isoprenaline, which correlate with ligand efficacy (Fig. 5; Supplementary Fig. 9), indicate conformational variations among the ternary complexes relating to the specific orthosteric ligand bound. A loss in relative signal intensities suggests increased exchange broadening at lower temperature (Supplementary Table 6) and along with the observed upfield shifts at 293 K (Supplementary Figs. 7, 10; Supplementary Note 3) and the ligand efficacy correlation, indicates the presence of a

two-state equilibrium for ternary complexes in fast exchange on the NMR timescale at 308 K (Supplementary Table 3).

Two mechanisms have been proposed to explain ligand-dependent efficacy: partial agonists may cause a distinct ternary complex structural state, with reduced activity compared to the full agonist[43]; alternatively, ligand efficacy may modulate an equilibrium between inactive and fully active ternary forms of the receptor[44]. Our observations suggest that ligand efficacy modulates both the equilibrium between inactive and pre-activated states, and the conformational equilibrium between a fully activated receptor ternary complex ($A^{G+}$) and a less active ternary state ($A^{G-}$), $A^{G-} \rightleftharpoons A^{G+}$ (Fig. 8). The two states likely differ in the ability of the cytoplasmic region to interact with IBPs, resulting in a variable response. As ligand efficacy increases, the receptor samples more of the fully activated ($A^{G+}$) form, with adrenaline the most active ligand in our study. This is similar to the $A_{2A}R$; however, no structural changes are observed between the pre-active state and the ternary complex possibly due to the use of a $G\alpha_s$ C-terminal peptide fragment or because $A_{2A}R$ adopts an almost fully active state in the absence of IBP[19]. As the study uses only one reporter ($V229C^{6.31}$), structural changes distant from this site remain undetected. Ligand efficacy correlations are observed across the cytoplasmic and extracellular halves of $\beta_1AR$, indicating that the $A^{G-} \rightleftharpoons A^{G+}$ equilibrium correlates with global changes in the receptor conformation (Fig. 5), which are in fast exchange on the NMR timescale (Supplementary Table 3). Relative signal intensities for $M223^{5.54}$ and $M296^{6.41}$ show that all ternary complexes are more rigid than the other receptor states in this study, consistent with receptor stabilisation on coupling to an IBP. The cyanopindolol-bound ternary complex remains slightly more mobile (~80% maximum intensity for $M223^{5.54}$), consistent with its lower efficacy as it preferentially samples the less-active $A^{G-}$ state (Fig. 2; Supplementary Figs. 4, 6). In all ternary complexes, $M296^{6.41}$ shows more exchange broadening than $M223^{5.54}$, indicating residual TM6 motion (Supplementary Fig. 4).

$\beta AR$ are reported to show basal activity when bound to IBPs[5]. For the apo state, we measured nanobody affinity in the μM range, substantially reduced from the nM affinity observed in ternary complexes[38, 41]. Adding an excess of nanobody demonstrated that nanobody-bound ligand-free receptor spectra have strong similarities to ternary nanobody complexes, consistent with an active state (Fig. 6; Supplementary Fig. 11).

Spectral differences between the basal form and the ternary complexes are small compared to the (unliganded) apo form, and together with the lack of substantial changes in the cytoplasmic residues M153 (IL2) and $M283^{6.28}$ suggest that the main structural changes in TM6 and the cytoplasmic region occur on nanobody binding (Supplementary Fig. 11). This is endorsed by the normalised signal intensities of $M223^{5.54}$ and $M296^{6.41}$, which show more restricted dynamics for the basal $\beta_1AR$–nanobody complexes, closer to the ternary complexes than the apo receptor (Fig. 2). It also suggests that ligand binding causes further tightening of the receptor on ternary complex formation.

Isoprenaline addition to the preformed nanobody-bound ligand-free receptor results in identical spectra to the ternary complex formed upon nanobody addition to the isoprenaline-bound receptor (Supplementary Fig. 12). Similar observations are made for the other agonists (cyanopindolol, salbutamol and adrenaline) indicating that the final conformation of the ternary complex is independent of the binding order.

Nanobody complex formation and spectral similarities between the basal state and the ternary complexes suggest that sampling of the pre-activated (A) state by the apo form of $\beta_1AR$ without an agonist[20, 22] can allow weak coupling to G-protein with the potential for downstream signalling, providing an explanation for

the basal activity. $M223^{5.54}$ and $M296^{6.41}$ chemical shifts in the basal state correlate with the lowest-efficacy level for nanobody-bound complexes investigated (Fig. 5), suggesting that the basal state corresponds to the minimally activated ($A^{G-}$) complex in the $A^{G-} \rightleftharpoons A^{G+}$ equilibrium. Given the existence of ligand-free basal-state GPCR-G protein complexes, and the flexibility of the (A) state, receptor selectivity for intracellular signalling pathways may, in part, be influenced by which IBPs are expressed and present at the cell membrane in different cell types, potentially explaining the ability of a given ligand to activate multiple signalling pathways.

Spectra for the thermostabilising M90A mutant ($\beta_1AR$-Met$\Delta$5-M90A)[39] in the apo state are hardly changed from $\beta_1AR$-Met$\Delta$5 (Supplementary Fig. 13), indicating that in the absence of a ligand, the thermostabilising mutation has little effect. The shift of $M223^{5.54}$ and $M296^{6.41}$ signals in ligand-bound spectra of $\beta_1AR$-Met$\Delta$5-M90A towards those from $\beta_1AR$-Met$\Delta$5 when bound to lower-efficacy ligands, both in the absence and presence of Nb6B9 (Fig. 7a, c), implies that the thermostabilising mutation reduces receptor activity. Temperature-dependent chemical shift changes of similar size and sign to $\beta_1AR$-Met$\Delta$5 are still observed for $M223^{5.54}$ and $M296^{6.41}$, emphasising that the fast-exchanging equilibria, $I \rightleftharpoons A$ and $A^{G-} \rightleftharpoons A^{G+}$, still exist, albeit shifted towards the less active state for each equilibrium (Supplementary Fig. 7).

Relative peak intensities show dramatically reduced μs-to-ms dynamics for isoprenaline- and salbutamol-bound $\beta_1AR$-Met$\Delta$5-M90A, consistent with a shift in the equilibria, revealing substantially more rigid agonist-bound receptor (Fig. 7a, b), while the dynamics of the apo form is hardly affected. The relative signal intensities do not show any temperature or field dependence, indicating reduced sampling of the (A) state via a left-shift in the $I \rightleftharpoons A$ equilibrium (Supplementary Table 7). Based on our data, it is currently not clear if stabilisation also changes the dynamics of the (A) state, which might change the sampling of the active-like (A′, A″, A‴…) conformations leading to sharpening of the $M223^{5.54}$ and $M296^{6.41}$ signals or if the (A) state is just less populated. As the receptor retains its ability to form ternary complexes, it can be assumed that such states are being sampled. The ternary complexes show no change in dynamics compared to $\beta_1AR$-Met$\Delta$5. Overall, thermostabilisation is similar to reducing the efficacy of a particular ligand, shifting ligand-bound and ternary complex equilibria towards the less active state.

Using NMR spectroscopy, we investigated the conformational diversity and dynamic nature of $\beta_1AR$ activation through orthosteric ligands and $G_s$-mimetic nanobodies. We provide structural insight into partial agonism and basal activity, presenting our conclusions as a model (Fig. 8). The receptor populates two ligand efficacy-dependent equilibria, between an inactive (I) and pre-activated (A) form and, for the receptor bound to $G_s$-mimetic, between more ($A^{G+}$) and less ($A^{G-}$) active forms of the nanobody-bound receptor. The less active IBP-bound form is consistent with the conformation of the basal activity complex. Full agonist binding results in a highly dynamic pre-activated form that samples multiple active-like conformations, competent to bind IBPs. Two different conformations can be distinguished at lower temperature with additional broadened or low-populated states possibly remaining undetected. We suggest that the different states enable preferential interaction with particular IBPs with the relative populations of these states modulated by different ligands. Bound to a $G_s$-mimetic, the receptor becomes less mobile with allosteric modulation of the cytoplasmic interaction by the ligand in the orthosteric binding pocket. According to our data, orthosteric ligand efficacy is converted into a distinct conformational response on the cytoplasmic side. Differences between the ($A^{G-}$) and ($A^{G+}$) states occur on TM5 and TM6, indicating that these conformational variations may modulate the

receptor–nanobody interaction. We speculate that when $\beta_1AR$ is bound to $G\alpha_s$, these differences may contribute to conformational changes regulating the rate of GDP release, providing control of the GEF function of the GPCR. Through this mechanism, ligands may modulate the efficacy of downstream signalling by the receptor.

## Methods

**Constructs and mutagenesis**. The $\beta_1$AR-Met2 construct used here was derived from the truncated turkey $\beta_1$AR receptor $\beta44$-m23[34]; through the reversal of three stabilising mutations (V90M[2.53], A227Y[5.58] and L282A (IL3)), and the inclusion of two further mutations (E130W[3.41] and D322X[7.32]). To reduce spectral overlap, five methionine residues were mutated (M44L[1.35], M48L[1.57], M179L (EL2), M281A (IL3), M338A[7.48]) along with a further reversal of the stabilising mutation (K322D[7.32]), creating the construct Met2-Δ5. Therefore, Met2-Δ5 differs from the wild-type receptor in the truncation of the N- and C-termini, and of intracellular loop 3 (IL3), in addition to the presence of four thermostabilising point mutations, the mutation of five methionine residues as well as the presence of a mutation increasing expression yield (C116L[3.27]) and the removal of a palmitoylation site (C358A). For ease of purification, a C-terminal octahistidine tag was added. The sequence of Met2-Δ5 is.

MGAELLSQQWEA GLSLLLALVVLLI VAGNVLVIAAIGSTQRLQTLTNLFIT SLACADLVMGLLVVPFGA TLVVRGTWL WGSFLCELWTSL DVLCVTASIWT LCVIAIDRYLAITS PFRYQSLM TRARAKVIICTVW AISALVSFLPIMLHWWR DEDPQALKCYQDP GCCDFVTN RAYAIASSIISFYIPL LIMIFVYLRVYREAKE QIRKIDRASKR KTSRVAAMREH KALKTLGIIMGVFT LCWLPFFLVNIVNVFN RDLVPDWLFVAFNWLGY ANSAANP IIYCRSPDF RKAFKRLLAFPRKAD RRLHHHH HHHH.

For assignment purposes, methionine residues were substituted using site-directed mutagenesis in a PCR reaction (primer sequences given in Supplementary Table 10) using Phusion DNA polymerase (ThermoFisher). *DpnI*-digested PCR product was transformed into DH5α *Escherichia coli* cells (New England Biolabs) along with the transfer plasmid, and the resulting plasmid DNA construct was isolated using a commercial Miniprep kit (Qiagen) ready for transfection in insect cells.

**Expression and purification of $\beta_1$AR**. Baculovirus was generated for expression using FlashBac viral DNA (Oxford Expression Technologies). $\beta_1$AR containing pBacPAK8 plasmid was diluted to 100 ng μL$^{-1}$ and 1.8 μL, together with 1.8 μl of FlashBac DNA, was added to 1.8 μl of previously dissolved and NaOH-neutralised polyethylenimine (linear PEI 25000, Polysciences, 1 mg mL$^{-1}$ concentration) and diluted with 360 μL of cell culture media (SF4, Bioconcept). The mixture was incubated at room temperature for 45 min to allow for DNA–polymer complexes to form. The complex mixture was added to 1 mL of mid-log phase Sf9 cells (ThermoFisher), diluted to $0.5 \times 10^6$ cells mL$^{-1}$ in serum-free SF4 media and incubated at 27 °C shaking for 5 days. On day 5, a small sample was taken to confirm visual signs of infection. The resulting P0 viral stock was diluted to 4 mL with fresh media and incubated for 48 h to generate P1 viral stock. An aliquot of 100 μL of P1 stock was used to infect 50 mL of cells at a density of $1 \times 10^6$ cells mL$^{-1}$, and incubated at 27 °C shaking for 48 h, generating a high-titre P2 stock for protein expression[45].

For $\beta_1$AR expression, Sf9 or Sf21 cells (ThermoFisher), grown in serum-free SF4 media were centrifuged (500×g, 10 min) and washed with sterile PBS, to reduce the carry-over of unlabelled methionine. The washed cells were diluted to a density of $3 \times 10^6$ cells mL$^{-1}$ into methionine-deficient SF4 media at half the intended final culture volume. The culture was then infected with 4 mL L$^{-1}$ of high-density viral stock, and incubated for 5 h, before supplementing the culture with 250 mg L$^{-1}$ of $^{13}$C methyl methionine and diluting to a final density of $1.5 \times 10^6$ cells mL$^{-1}$. The initial reduction in culture volume ensures optimal aeration in the initial phase of the viral infection. Cells were grown at 27 °C for 48 h and were harvested by centrifugation (3500×g, 15 min).

The frozen insect cell pellet was thawed with solubilisation buffer (20 mM Tris-HCl, pH 8.0, 350 mM NaCl, 3 mM imidazole, Complete Protease Inhbitor Cocktail (Roche) and 2% LMNG) and stirred for 1 h. The solubilised cells were clarified by centrifugation (175,000×g, 45 min) and the soluble fraction was loaded onto a nickel affinity column. The column was washed with equilibration buffer (20 mM Tris-HCl, pH 8.0, 350 mM NaCl, 3 mM imidazole and 0.02% LMNG) and the protein was eluted with the same buffer supplemented with 250 mM imidazole. The final sample was exchanged into 10 mM Tris-HCl, pH 8.0, 150 mM NaCl and 0.04% LMNG.

**E. coli expression and purification of Nb80 and Nb6B9**. Nb80- and Nb6b9-containing plasmids were transformed into BL21(DE3) (New England Biolabs) cells and grown in LB media supplemented with the relevant antibiotics. Expression cultures were inoculated with a saturated overnight culture to an OD$_{600}$ density of 0.1 AU and grown to a density of 0.8 AU at 37 °C. Induction was achieved with isopropyl thiogalactopyranoside (IPTG) at a final concentration of 0.5 mM. Expression cultures were grown at 25 °C for 16 h before harvesting by centrifugation (3500×g, 20 min, 4 °C).

Frozen cell pellets were thawed in 20 mM Tris-HCl, pH 8.0, 150 mM NaCl with Complete Protease Inhibitor Cocktail (Roche) and lysed with an EmulsiFlex C5 homogeniser (Avestin). The solubilised cells were clarified by centrifugation (175,000×g, 15 min) and the soluble fraction was loaded onto a nickel affinity column. Unbound sample was removed with wash buffer (20 mM Tris-HCl, pH 8.0, 150 mM NaCl), followed by a further wash of the same composition supplemented with 6 mM imidazole. Nanobodies were eluted with wash buffer containing 250 mM imidazole. A further size-exclusion chromatography step on a Superdex S200 10/300 Increase column yielded a 95% pure protein preparation in 20 mM Tris-HCl, pH 8.0, 150 mM NaCl. Typical yields of Nb80 and Nb6b9 obtained by this method were 25 and 4 mg L$^{-1}$, respectively.

**NMR experiments**. NMR samples were prepared with 5% D$_2$O and the ligand was added directly to the receptor in the apo form. The final ligand concentrations were 1 mM salbutamol, isoprenaline, adrenaline, 200 μM carvedilol, 140 μM or 600 μM S (−)-cyanopindolol and 100 μM 7-methylcyanopindolol. Ligand-bound populations were all > 99.9% (Supplementary Table 11). Nanobody was added in a molar excess for ternary complex formation and in 15-fold excess for basal complex formation. NMR experiments were recorded on a Bruker Avance AVIII 800 spectrometer ($^1$H 800 MHz) equipped with a 5-mm TXI HCN/z cryoprobe or where specified on a Bruker Avance AVIII 600 spectrometer ($^1$H 600 MHz) equipped with a 5-mm QCI HCNF/z cryoprobe.

Unless specifically mentioned experiments were acquired at 308 K using a SOFAST $^1$H, $^{13}$C HMQC experiment[46] with gradient coherence-order selection[47] and non-uniform sampling (NUS) in the $^{13}$C dimension. Gradient selection was required to reduce the intense LMNG detergent signals, while selective excitation of methyl groups enabled use of a short recycle delay of 0.5 s. Excitation used a 2.25-ms 120° PC9 pulse[48] and inversion a 1.16-ms, 180° REBURP pulse[49]. A 60% Poisson-gap sampling schedule was used (60 complex points from a total of 100 complex points)[50], with a maximum acquisition time ($t_{max}$) of 25 ms and spectral width ($^{13}$C) of 4000 Hz. The direct ($^1$H) dimension was acquired with 10,000 Hz spectral width, 1024 points and $t_{max}$ = 51.2 ms. Spectra were recorded with 368 scans, giving an acquisition time of ~6 h. Where higher sensitivity was required, multiple 6 h experiments were recorded and added. Spectra were reconstructed using the iterative hard thresholding (IHT) compressed sensing (CS) implementation[51] in the Cambridge CS package (M.J. Bostock, unpublished). Data were analysed using CCPN Analysis v2[52].

**Data availability**. The authors declare that relevant data supporting the findings of this study are available within the article and its Supplementary Information files or on reasonable request from the corresponding author. NMR shifts are available at the BioMagResBank (accession numbers 27292–27297).

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

## Acknowledgements

This work was funded in parts through a BBSRC research grant to D.N. (BB/K01983 X/1) and studentship support to A.S. (MRC Industrial CASE).

## Author contributions

A.S., M.J.B. and D.N. designed the research. A.S. performed molecular biology, protein expression and purification. Protein expression was carried out in collaboration with B.S. P.K. helped with molecular biology and construct design. T.W. and C.G.T. provided initial receptor and nanobody constructs and assisted with receptor biochemistry. A.S., M.J.B. and D.N. carried out NMR data collection, data processing, spectral assignments and analysed the data. M.J.B., A.S. and D.N. prepared the manuscript. D.N. supervised the project.

## Additional information

**Competing interests:** The authors declare no competing financial interests.

