## [Peer Review File · Nature Communications]

Reviewers' comments:

Reviewer #1 (Remarks to the Author):

In this paper, the authors make use of ¹³C-methionine and methyl spectroscopy to identify conformational equilibria in the beta-1 adrenergic receptor. The work represents a significant advance and should definitely be published. This was made possible by cleverly tailoring the methionine reporters so as to avoid spectral overlap while making use of a minimum of thermostabilizing mutants. There are two important findings in the paper regarding the observation of: 1) allostery between the cytoplasmic and extracellular domains of the receptor and the effects of either ligand or nanobody and 2) the equilibrium between a preactive and active state which is directly influenced by partial agonist and agonist respectively. The latter finding is intriguingly advanced through the observation of dynamics in this equilibrium and the correlation between dynamics and activity.

1. The notion of dynamics and the relationship to function is very eloquently discussed and treated in this paper. Given issues of sample stability, the authors resort to a "poor man's" surrogate of relaxation experiments - namely normalized peak height. A peak height could drop for many reasons in addition to those discussed by the authors. Anything that changes T₂ (e.g. packing and a change in dipolar relaxation terms, a change in orientation and CSA relaxation...) could change peak height. Given the extensive number of drugs considered I'm totally in agreement with the authors conclusions though they may wish to add a sentence cautioning readers on this approach.

In an earlier study by Kim et al (JACS, 2013, 135, 9465) the authors made use of field dependent T₁ and T₂ to evaluate relative dynamics of inactive and active states. The work was more limited in scope as only a single site was labeled and studied. However, the active state was promoted by agonist and observed to be the most dynamic in terms of amplitude. This work should be referenced as it shows that it is possible to perform these types of relaxation experiments in GPCRs.

2. In the discussion the authors begin to talk about substates (A, A', ...) and write "These states may couple to different IBPs, suggesting that the ability of the receptor to access different conformations in the pre-active state plays an important role in engaging different downstream signalling pathways." They should replace "suggesting" with "in which case". This is fine to speculate about if written as such.

3. The paper by Ye et al (Nature 533, 265–268 (2016)), referenced by the authors, puts forward exactly the same idea regarding a G⁻ to G⁺ equilibrium. The strength of the current paper is in the number of reporters such that allostery and overall tertiary structure in response to drug can in some way be assessed. The authors suggest that the work of Ye showed no change in structure of the preactive state upon binding of a G_s peptide and that this might be due to the limited contact area resulting from the peptide. I agree. However, the study by Ye et al involved a single reporter so it is also difficult to say what the receptor "looks like" far away from the reporter. For the sake of the readers, the authors should probably acknowledge this as well.

4. Stylistically, I like the figures but I feel they need to be more annotated for readers to understand.

5. The authors should probably invoke limits on exchange rates based on maximal chemical shift differences seen in Hz at low temperature and ranges of shifts for certain equilibria. It's important because what a ¹⁹F NMR person calls slow is what a ¹³C,¹H NMR person calls intermediate, given chemical shift dispersions as they are.

6. I like Figure 8 and the model. We have the same observation in our system which is exciting to me. Is it still possible that G⁻ is on pathway to G⁺ and that partial agonist simply stabilizes G⁻ or

are there definitely 2 activation pathways?

Reviewer #2 (Remarks to the Author):

The authors have used NMR to investigate the conformational diversity and dynamic nature of coupling between activated receptor states of turkey β 1AR and Gs mimetic nanobodies. The study nicely complements and extends the surge of crystallographic data towards an understanding of the structural dynamics in receptor activation and (biased) signalling by providing insights into structural changes related to ligand binding (extending beyond the ligand-binding pocket), increased dynamics of the agonist bound state, structural changes of ligand-only and ternary complex, and to my knowledge the first demonstration of this approach to study the effect of a thermostabilising mutation. Hence, the article is of sufficient novelty and value to motivation publication.

Some comparison is done to the homologous β 2AR prototype. However, it would be nice to broaden the discussion on the applicability of the results towards the crystallised or all class A GPCRs by providing statistics on conservation of key studied residues, ideally using structure-based sequence alignments (to avoid for offsets due to helix bulges and constrictions). Another way would also be to compare to other published studies where the structural interplay in pharmacology of IBP and ligands have been analysed with nanobody mimics, such as: Staus, D. P. et al. Allosteric nanobodies reveal the dynamic range and diverse mechanisms of G-protein-coupled receptor activation. Nature 535, 448-452.

For future studies, it would be interesting to combine the NMR approach with molecular dynamics simulations when the computational power can allow for sufficient time scales. Pragmatic approximations can be done already now by for example removing a IBP (mimic) or exchanging two ligands (with different mode of action or bias). For completion and reference, the authors should cite at least one such study.

Reviewer #3 (Remarks to the Author):

In the manuscript 'Insight into partial agonism by observing ligand-modulated conformational equilibria of a Gs-mimetic nanobody-bound β 1-adrenergic receptor' Solt and colleagues experimentally describe the affects of a selective partial and full agonist, an antagonist, and two nanobodies on the activation of the β 1-adrenergic receptor when compared to ligand free (apo) as they relate to the equilibria and movement of key regions within the receptor. The experiment was accomplished using NMR and methionine and carbon labeled residues positioned within the transmembrane region along with the intracellular and extracellular loop(s) of the receptor. My overall impression of the manuscript is that it is well written with no major flaws in the NMR data analysis as presented, data supports the ternary complex model of GPCR activation and provides insight in ligand selectivity. My concerns are as follows:

- 1) Title – the title of the manuscript is misleading as the authors describe multiple states of ligand equilibria and not just a partial agonist. Revise the title to better describe the entire manuscript.
- 2) Previous crystal structures of β 1-adrenergic receptor have been determined

but no comparison was made to these structures to support or expand on their NMR analysis. The authors should compare structures of the ligand bound states to help further support their claims.

3) Authors should explain how the thermostabilized mutant, M90A, was identified, ie. which ligand was it identified against, this would imply that thermostabilizing against a given ligand can not be universally applied to all ligands.

4) Figure 8 should expand on the entire ternary complex theory, the authors only describe the activated state, t/- g-protein or mimetic

5) Not described in the methods section is the concentration of ligands using in establishing the transition states. Also not thoroughly discussed or adequately referenced was an interpretation or description of ligand on and off rates and how this would affect the NMR spectra when interpreting the transition states.

Reviewer #1:

We would like to thank the reviewer for his/her favourable reaction and enthusiasm to our manuscript and for the assessment of allostery between the cytoplasmic and extracellular domains and the equilibrium between preactive and active states as essential developments of this work. In the following we respond to the reviewer's points.

1. *The notion of dynamics and the relationship to function is very eloquently discussed and treated in this paper. Given issues of sample stability, the authors resort to a "poor man's" surrogate of relaxation experiments - namely normalized peak height. A peak height could drop for many reasons in addition to those discussed by the authors. Anything that changes T2 (e.g. packing and a change in dipolar relaxation terms, a change in orientation and CSA relaxation...) could change peak height. Given the extensive number of drugs considered I'm totally in agreement with the authors conclusions though they **may wish to add a sentence cautioning readers on this approach.***

We agree with the reviewer that whilst our data supports the conclusions drawn, it is important to make the reader aware of other possible contributions, which could affect peak height. We have added two additional sentences on pg 6 to emphasise this.

In an earlier study by Kim et al (JACS, 2013, 135, 9465) the authors made use of field dependent T1 and T2 to evaluate relative dynamics of inactive and active states. The work was more limited in scope as only a single site was labeled and studied. However, the active state was promoted by agonist and observed to be the most dynamic in terms of amplitude. This work should be referenced as it shows that it is possible to perform these types of relaxation experiments in GPCRs.

We have included this additional reference on pg 5 as requested by the reviewer to make readers aware of the possibility of performing such measurements on GPCRs under some, favourable, circumstances.

2. *In the discussion the authors begin to talk about substates (A, A', ...) and write "These states may couple to different IBPs, suggesting that the ability of the receptor to access different conformations in the pre-active state plays an important role in engaging different downstream signalling pathways." They should **replace "suggesting" with "in which case"**. This is fine to speculate about if written as such.*

We have changed this sentence on pg 9 as requested by the reviewer.

3. *The paper by Ye et al (Nature 533, 265–268 (2016)), referenced by the authors, puts forward exactly the same idea regarding a G- to G+ equilibrium. The strength of the current paper is in the number of reporters such that allostery and overall tertiary structure in response to drug can in some way be assessed. The authors suggest that*

the work of Ye showed no change in structure of the preactive state upon binding of a Gs peptide and that this might be due to the limited contact area resulting from the peptide. I agree. However, the study by Ye et al involved a single reporter so it is also difficult to say what the receptor “looks like” far away from the reporter. For the sake of the readers, the authors should probably acknowledge this as well.

We agree with the reviewer that it is important to make readers aware of the differences in experimental techniques and what can be concluded from them between these two papers (e.g. the number of reporters used) and have added an additional sentence on pg 11 as requested by the reviewer.

4. Stylistically, I like the figures but I feel they need to be more annotated for readers to understand.

We agree with the reviewer that it is important for the figures to be well annotated to make it easy for the reader to comprehend the underlying message. The comment does not specify particular figures; in preparing the figures we endeavoured always to make these as clear as possible to the reader. We have nevertheless added some additional annotations/alterd the labelling in order to make sure the figures are clear as listed below:

Figure 1:

Sub-figure titles now in bold so these can be clearly distinguished.
(h) the key is now clearly labelled as increasing efficacy.

Figure 2:

An arrow has been added to make clear the direction of increasing dynamics.

Figure 4:

The key on the figure now says +Isoprenaline and +Isoprenaline+Nb6B9 to make it clearer which conditions are shown.

Figure 5:

(a) Legend now says +Nb6B9 and +Nb6B9+Isoprenaline to clarify the conditions shown.
(k) The key is now clearly labelled as ‘efficacy’.

Figure 6:

The key has been adjusted to make this figure clearer.

Figure 7:

(a) and (c) The keys have been reorganised to make this clearer.
(b) An arrow has been added to emphasize the direction of increasing dynamics.

Figure 8:

This figure has been substantially modified. See Reviewer 3 #4.

We have also made a number of modifications to the Figure legends (highlighted in red) with the hope of making it easier to interpret the figures.

5. The authors should probably invoke limits on exchange rates based on maximal chemical shift differences seen in Hz at low temperature and ranges of shifts for certain equilibria. It's important because what a ¹⁹F NMR person calls slow is what a ¹³C,¹H NMR person calls intermediate, given chemical shift dispersions as they are.

We agree with the reviewer that this information is necessary to enable comparisons with other papers and have included additional information on exchange rates in the supplementary information (pg 1&2) with rates calculated for the four type of different equilibria studied in this paper shown in Supplementary Table 3.

6. I like Figure 8 and the model. We have the same observation in our system which is exciting to me. Is it still possible that G⁻ is on pathway to G⁺ and that partial agonist simply stabilizes G⁻ or are there definitely 2 activation pathways?

This is an important question which we have also considered. As observed by ourselves (and others), the (A) state is a highly dynamic state, which we represent in our model as A', A'' etc. It is highly likely that a number of the dynamic conformations which make up the A state are able to couple to G_s with varying affinities, whilst other conformations couple to other downstream signalling molecules via alternative signalling pathways. This coupling could lead to formation of A^{G-} or A^{G+} states. It is also possible, as the reviewer notes, that coupling to G_s leads to formation of A^{G-} which is on route to formation of A^{G+}. We think it is most likely that both paths are possible, however, it is not as yet possible for us to distinguish between these two hypotheses. We have adjusted Figure 8 so that it is clear that this question remains ambiguous.

Reviewer #2:

We wish to thank the reviewer for his/her helpful suggestions and for the recognition of the novelty of our manuscript. Specific points are addressed below.

1. *Some comparison is done to the homologous β 2AR prototype. However, it would be nice to broaden the discussion on the applicability of the results towards the crystallised or all class A GPCRs by providing statistics on conservation of key studied residues, ideally using structure-based sequence alignments (to avoid for offsets due to helix bulges and constrictions). Another, way would also be to compare to other published studies where the structural interplay in pharmacology of IBP and ligands have been analysed with nanobody mimics, such as: Staus, D. P. et al. Allosteric nanobodies reveal the dynamic range and diverse mechanisms of G-protein-coupled receptor activation. Nature 535, 448-452.*

We would like to thank the reviewer for this suggestion. We agree that as well as the mechanistic insight into the β -adrenergic receptor function provided by our manuscript, the study also provides important general principles addressing the more general mechanisms underpinning GPCR function. We have made a number of comparisons throughout the text to other receptors particularly to structures of the β_2 AR (pg 8-11 and Supplementary Discussion) as well as to the general mechanism underpinning ligand-dependent efficacy in GPCRs (pg 11, paragraph 2) with comparisons to the context of the A_{2A} receptor. This is summarised in our proposed model in Figure 8. Our study relies on the use of specific reporters placed in different, structurally important parts of the receptor. This is comparable to other NMR approaches which have variously used methionines, tryptophans or single fluorescent tags attached to cysteines. While methyl groups provide particular spectroscopic properties, which make them favourable as reporters (Supplementary Discussion), in line with other similar NMR studies, the reporters are used as proxies to assess the structural and dynamic changes occurring at particular sites in the GPCR.

We have included an additional supplementary table (Supplementary Table 1) comparing the reporter positions in our turkey β_1 AR construct with other class A GPCRs (human β_1 , β_2 and β_3 , and human A_{2A}), as well as information on the most common residue at each position, the % of class A receptors with Met at a given position, and the position of Met in the rank order of most common residues. This shows that for the key sites discussed in our paper (M90, M153, M223 and M296) Met is at least the 3rd most common residue.

Nevertheless, we would like to emphasise that the central idea underpinning our approach is that such residues can be used as reporters of the underlying structural and dynamic changes occurring in the β_1 AR and so the identity of the residue is of secondary importance.

- 2. For future studies, it would be interesting to combine the NMR approach with molecular dynamics simulations when the computational power can allow for sufficient time scales. Pragmatic approximations can be done already now by for example removing a IBP (mimic) or exchanging two ligands (with different mode of action or bias). For completion and reference, the authors should cite at least one such study.*

We agree with the reviewer that MD simulations are likely to make important contributions to this area in the future, in particular as computing power allows increased timescales. We have added an additional sentence and reference (Dror, R. O. *et al.* Activation mechanism of the β_2 -adrenergic receptor. *Proc. Natl. Acad. Sci.* **108**, 18684–18689 (2011)) on pg 3 outlining the importance of such approaches.

Reviewer #3 (Remarks to the Author):

We would like to thank the reviewer for his/her comments. Our responses are below.

1. *Title – the title of the manuscript is misleading as the authors describe multiple states of ligand equilibria and not just a partial agonist. Revise the title to better describe the entire manuscript.*

We thank the reviewer for his/her careful consideration for our title. The title we have chosen emphasises the central point of the paper, which is that partial agonism can be understood as modulation of the various equilibria which are involved in the activation pathway of GPCRs. We identify two equilibria; between inactive and preactivated receptor and between less and more activated G-protein nanobody complexes. The efficacy of the ligand determines the position of the equilibrium in both cases. Basal activity can be understood as an extreme of these equilibria, representing the left hand side (low activity) of the $A^{G^-} \rightleftharpoons A^{G^+}$ equilibrium, while the thermostabilising mutant (M90A, see response to point 3) represents a shift of both of these equilibria to the left hand side (lower activity). Our original title aimed to reflect these central observations. In response to the reviewer's comment, we have slightly altered the title to make it clear that we observe multiple equilibria which represent both ligand binding and nanobody binding. We trust that our revised title, "*Insight into partial agonism by observing multiple equilibria for ligand-bound and G_s -mimetic nanobody-bound β_1 -adrenergic receptor*" is clearer.

2. *Previous crystal structures of β_1 -adrenergic receptor have been determined but no comparison was made to these structures to support or expand on their NMR analysis. The authors should compare structures of the ligand bound states to help further support their claims.*

As the reviewer notes, a number of crystal structures of the β_1 AR coupled to various ligands have been reported (e.g. Warne, T. *et al.* The structural basis for agonist and partial agonist action on a β_1 -adrenergic receptor. *Nature* **469**, 241–244 (2011)). However, as discussed in the supplementary information (pgs 4 & 5 and Supplementary Table 8), the constructs used in these studies contain the thermostabilising Y227A^{5,58} mutation which is known to impair coupling to intracellular binding partners (IBPs) e.g. nanobodies and G_s . In addition a number of the β_1 AR structures also contain Y343L^{7,53}, which also impairs coupling. In contrast, our construct does not contain these mutations. As result crystal structures of β_1 AR bound to several different ligands (partial agonists salbutamol, and dobutamine; full agonists carmoterol and isoprenaline) show changes concentrated around the ligand binding pocket, with no changes in the cytoplasmic region of the receptor. Consequently, these structures are believed to represent a thermostabilised, inactive state of the receptor, with no substantial changes compared to antagonist bound receptor, aside from the region of the ligand binding pocket. In contrast, our construct shows changes in response to ligand binding across the receptor and particularly in the cytoplasmic region (helices 5 and 6). As a result, for a fair comparison of our data with existing crystal structure information, and due to the lack of β_1 AR crystal structures in the presence of nanobody or G protein, we have made comparisons to structures of the β_2 AR in the supplementary discussion (pgs 4-9 and Supplementary Figure 14). We believe this comparison is the most relevant due to the substantial changes observed in the β_2 AR on coupling to IBPs.

In order to clarify our approach to the reader, we have added additional information on the β_1 AR crystal structures on pg 5 of the Supplementary information highlighting the limited

changes observed in the existing β_1 AR crystal structures and further explaining the choice of comparisons to the β_2 AR. We have also added a sentence referring to this in the main text on pg 10. We thank the reviewer for his/her identification of this important point, and trust that our logic is now clearer to the reader.

- 3. Authors should explain how the thermostabilized mutant, M90A, was identified, ie. which ligand was it identified against, this would imply that thermostabilizing against a given ligand can not be universally applied to all ligands.*

The conformationally thermostabilising mutant 90A was identified in previous work against the antagonist alprenolol as described already on pg 8 and reference 39. This mutant is reported to increase the T_m for the alprenolol bound state of β_1 AR₃₄₋₃₂₄ by +8 °C (Serrano-Vega, M. J., Magnani, F., Shibata, Y. & Tate, C. G. Conformational thermostabilization of the β_1 -adrenergic receptor in a detergent-resistant form. *Proc. Natl. Acad. Sci. U. S. A.* **105**, 877–882 (2008)). Receptor conformational thermostabilisation can manifest as both thermodynamic and kinetic stabilisation. The reported increase in T_m (+8 °C) against alprenolol suggest a thermodynamic stabilisation as we can see in the NMR spectra that the equilibrium is shifted slightly towards a less active state. We are unable to comment on the type of stabilisation in the presence of other ligands, however, our NMR data suggests the occurrence of kinetic stabilisation in the presence of salbutamol and isoprenaline ligands (Figure 7) where conformational exchange is slowed. In Figure 7a. At the same time M296 peaks for the M90A mutant are shifted towards the less active state for both isoprenaline and salbutamol-bound receptor compared to the equivalent peaks in the $\Delta 5$ construct. This suggests lower activity of the M90A mutant, which could correspond to increased thermodynamic stabilisation of the mutant bound to these ligands; this is further supported by the increased relative peak intensities for M90A apo, salbutamol-bound and isoprenaline-bound receptor (Figure 7b) relative to the $\Delta 5$ construct. The principle of conformational thermostabilisation in selecting for stabilising effects (mutants) towards a particular type of ligand should be widely applicable for different ligands. However, it is not clear that sufficient stabilisation can be achieved to produce a receptor stabilised in the fully active state conformation. For β_1 AR for example this seems unlikely as our investigations indicate at a relatively flat energy landscape, where a stable fully active state can't be 'isolated'. In this context our study and contributions from others show clearly that active states are highly dynamic, indicating that generation of a stabilised fully active state receptor is unlikely (unless combined with coupling to IBPs). In this context, depending on the thermodynamic landscape of the receptor, mutagenesis may allow stabilisation of early activation states as e.g. observed for A_{2A} making the success of thermostabilisation dependent on the conformational energy landscape explored by the receptor (Lebon, G. *et al.* Agonist-bound adenosine A_{2A} receptor structures reveal common features of GPCR activation. *Nature* **474**, 521–525 (2011)). A number of computational approaches have aimed at predicting the site of mutations that are particularly thermostabilising, however, there is currently no availability of reliable algorithms that would enable successful predictions on a wider scale.

4. *Figure 8 should expand on the entire ternary complex theory, the authors only describe the activated state, t/- g-protein or mimetic*

We thank the reviewer for this suggestion. Our aim with Figure 8 was to show the key findings from this paper, hence the selected states from the extended ternary complex model. However, we agree with the reviewer that it could be useful to the reader to include a more complete model and include ligand-free states. As a result we have adjusted the model to include ligand free (I) and (A) states (using the nomenclature of our model) and the link between these states and the ligand-efficacy dependent equilibrium and the basal activity of the receptor. We have adjusted the Figure legend accordingly.

5. *Not described in the methods section is the concentration of ligands using in establishing the transition states. Also not thoroughly discussed or adequately referenced was an interpretation or description of ligand on and off rates and how this would affect the NMR spectra when interpreting the transition states.*

We thank the reviewer for this suggestion and have now included this important information on ligand concentrations used under ‘NMR Experiments’ in the methods section (pg 15). In addition we have included a table (Supplementary Table 10), which shows that all ligands were added in excess leading to a ligand-bound population of >99.9%. Consequently, with receptor saturated with ligand we can assume a negligible contribution from unbound receptor, consistent with the observations in other similar NMR studies. Calculations for the exchange rate suggest the contribution to linewidth R_{ex} (in fast exchange) from the unbound population is smaller than 1 Hz throughout, compared to signal linewidths varying between 40 – 100 Hz .

In addition measurements for S(-)-cyanopindolol were conducted with concentrations of 600 μ M and 140 μ M (pg 16) and resulted in superimposable spectra. This is also consistent with observations made by Kofuku et al. (Kofuku, Y. *et al.* Efficacy of the β_2 -adrenergic receptor is determined by conformational equilibrium in the transmembrane region. *Nat. Commun.* **3**, 1045 (2012)).

We have also produced an additional Supplementary Table 3, where we show the exchange regime as well as from the NMR data derived upper/lower bounds for exchange rates k_{ex} for the different equilibria discussed in our work (see also reviewer 1 #5).